# PFSKANs: A Novel Pixel-Level Feature Selection Model Based on Kolmogorov–Arnold Networks

**DOI:** 10.3390/s25164982

**Published:** 2025-08-12

**Authors:** Rui Yang, Michael V. Basin, Guangzhe Yao, Hongzheng Zeng

**Affiliations:** 1Robotics Institute, Ningbo University of Technology, Ningbo 315211, China; yaoguangzhe@nbut.edu.cn; 2School of Physical and Mathematical Sciences, Autonomous University of Nuevo León, San Nicolás de Los Garza 66455, Nuevo León, Mexico; 3Key Laboratory of Flight Techniques and Flight Safety, CAAC, Guanghan 618307, China; zenghongzheng@cafuc.edu.cn

**Keywords:** feature selection, feature extraction, kolmogorov-arnold networks, pixel-level feature

## Abstract

**Highlights:**

**The main contributions and innovations of the proposed PFSKAN model are as follows:**
We propose an efficient, global, and interpretable feature extraction mechanism PFSKAN, which demonstrates superior performance over CNNs on low-resolution image classification datasets.A general pipeline for developing new feature extraction is provided, which utilizes the modified KAN interpretability technique to select key pixels, followed by pixel characteristic analysis to derive a generalized feature extractor or selector.A method for unifying the input dimension is proposed to address the issue of varying numbers of feature pixels across different images.

**Abstract:**

Inspired by the interpretability of Kolmogorov–Arnold Networks (KANs), a novel Pixel-level Feature Selection (PFS) model based on KANs (PFSKANs) is proposed as a fundamentally distinct alternative from trainable Convolutional Neural Networks (CNNs) and transformers in the computer vision tasks. We modify the simplification techniques of KANs to detect key pixels with high contribution scores directly at the input image. Specifically, a trainable selection procedure is intuitively visualized and performed only once, since the obtained interpretable pixels can subsequently be identified and dimensionally standardized using the proposed mathematical approach. Experiments on the image classification tasks using the MNIST, Fashion-MNIST, CIFAR-10, and CIFAR-100 datasets demonstrate that PFSKANs achieve comparable performance to CNNs in terms of accuracy, parameter efficiency, and training time.

## 1. Introduction

In image processing and computer vision tasks, the inherent nature of images often necessitates feature extraction and selection to scale down dimensionality, thereby enhancing the performance of downstream models. Feature extraction is more commonly employed to transform or encode raw pixels into effective visual representations. Conventional feature extraction methods are typically handcrafted, mainly including feature point detection, a histogram of oriented gradients, edge detection, local binary patterns, and Haar-like features [1]. They are advantageous in terms of strong interpretability and computational efficiency, consequently designed for specific visual tasks, such as image matching, face recognition, pedestrian detection, and texture analysis. Nevertheless, in generalized and complex vision tasks, these methods exhibit limited capacity to provide sufficiently expressive descriptors to support downstream models effectively. Deep learning-based feature extraction methods, such as Convolutional Neural Networks (CNNs) [2,3,4], autoencoders [5,6], and transformers [7,8,9,10], have become increasingly prevalent in recent years. These approaches demonstrate strong capabilities in addressing various complex visual tasks, including, but not limited to, classification, object detection, and image segmentation. However, they often involve a large number of parameters, require extensive training data, and generally lack interpretability.

Feature selection involves directly identifying and retaining the most informative components of the original feature space for downstream tasks. In computer visual tasks, it is common to first extract low-dimensional features from pixel-level input and then perform feature selection [11], which is increasingly integrated with deep learning approaches [12,13,14]. The primary advantage of feature selection lies in its capacity to eliminate the redundant and irrelevant features unmediatedly for improving performance [11]. It is particularly beneficial in cases where the data are high-dimensional but the number of available samples is limited [14], or interpretability is a critical requirement [15]. While selecting features from original input representations can yield more efficient dimensionality reduction, studies focusing on direct pixel-level feature selection remain scarce. This is largely due to the weak semantic features of individual pixels and the substantial computational cost associated with accurately identifying relevant pixels, regardless of whether filter-based, wrapper-based, embedded, or deep learning-integrated methods are employed.

Kolmogorov–Arnold Networks (KANs) represent a class of neural networks distinguished by their high nonlinearity and strong interpretability [16]. Employing spline functions along each edge, KANs are capable of achieving fitting performance comparable to, or even surpassing, that of multilayer perceptions (MLPs) while requiring significantly fewer parameters. Their interpretability is further enhanced through simplification techniques, which facilitate the identification of informative nodes based on activation values. The resulting pruned architectures allow for a more intuitive understanding of both intra-layer node relationships and inter-layer structural dependencies. Current research on KANs in the field of computer vision primarily focuses on adapting KANs by integrating CNNs and transformer-like components into their architecture, thereby broadening the range of their applicability [17,18,19,20,21,22,23,24]. Comparative studies between KANs and MLPs on various image-based tasks have demonstrated that KANs can achieve more compact model sizes and, in some cases, superior accuracy [25,26,27,28,29,30]. However, these advantages are accompanied by increased training time and reduced robustness to noise. Overall, KANs offer a promising framework for building compact, accurate, albeit with sensitivity to noisy data, and interpretable models with an inherent ability to selectively leverage input features.

The interpretability of KANs, in conjunction with their simplification techniques, enables unmediated feature selection at the pixel level. This capability offers two notable advantages. First, it facilitates highly efficient dimensionality reduction by identifying informative features across the entire spatial domain of the image, thereby mitigating the locality constraints and parameter inefficiencies typically observed in CNNs. Second, the inherent interpretability of KANs provides researchers with an intuitive understanding of the characteristics of the selected pixels, enabling the appropriate traditional feature extraction techniques to isolate relevant pixel-level features, significantly reducing computational cost. Combined with the highly nonlinear modeling capacity of KAN-based downstream networks, the overall parameter complexity can be further minimized, offering an effective alternative to transformer architectures, which often require substantial model size and large-scale training data.

The main contributions and innovations of the proposed PFSKAN model are as follows:We propose an efficient, global, and interpretable feature extraction mechanism PFSKAN, which demonstrates superior performance over CNNs on low-resolution image classification datasets.A general pipeline for developing new feature extraction is provided, which utilizes the modified KAN interpretability technique to select key pixels, followed by pixel characteristic analysis to derive a generalized feature extractor or selector.A method for unifying the input dimension is proposed to address the issue of varying numbers of feature pixels across different images.

## 2. Method

The PFS mechanism is developed to address two critical challenges in modern computer vision: the constrained receptive fields in CNNs that limit global feature extraction, and the prohibitive computational complexity of transformer-based approaches. As demonstrated in Figure 1, the proposed PFSKAN framework integrates these considerations through a novel architecture. Initially, interpretable feature selection is performed using the modified KANs’ analytical technique to extract high-contribution pixels for classification, with detailed methodology presented in Section 2.1. Subsequently, through systematic analysis of KAN-selected pixels, we propose the EdgeFuseTrim feature extractor that directly acquires pixel-level features across new datasets, bypassing the PFS generation process (yellow region in Figure 1), only requiring the PFSKAN’s workflow (purple region in Figure 1) for subsequent implementations. Detailed descriptions are provided in Section 2.2. Finally, we introduce a modified warping transformation to standardize input dimensionality, where the optimal dimension is determined through accuracy–dimensionality trade-off analysis, detailed in Section 2.3.

### 2.1. KAN-Based Key Pixel Selection

The pixel-level features in our framework refer to the key pixels identified using a modified interpretability technique based on KAN, which significantly contribute to classification performance. The selection process comprises the following three main steps: (1) training a complete KAN-based image classification model, (2) quantifying the contribution of each input pixel using a redesigned contribution score formula, and (3) determining an appropriate threshold to identify the key pixels. These steps are illustrated by the first three rectangular process blocks on the left yellow region in Figure 1.

To ensure precise quantification of input pixel contributions, it is essential to initially develop a KAN-based classification model exhibiting superior predictive performance. We conduct separate training of 2-layer KAN models on two benchmark datasets: (1) the MNIST dataset for grayscale image classification and (2) the CIFAR-10 dataset for color image classification, with their architectures detailed in the left of Figure 2. To incorporate spatial information of the image, the input layer is modified such that each pixel value is augmented by the sum of its corresponding x and y coordinates, defined as $I_{i,x}$ and $I_{i,y}$. As a result, the input dimensionality is doubled relative to the original image size. Then, all inputs are normalized to the range [−1, 1] using Min–Max Scaling. The exact mathematical formulation is provided in Equation (1), as follows:(1)$$\begin{array}{l} I_{i,x}=\left(I_{i}-I_{\min}\right)/\left(I_{\max}-I_{\min}\right)+x_{i}/x_{\max}-1\\ I_{i,y}=\left(I_{i}-I_{\min}\right)/\left(I_{\max}-I_{\min}\right)+y_{i}/y_{\max}-1 \end{array}$$
where $I_{i}$, $x_{i}$, and $y_{i}$ represent the intensity, x-coordinate, and y-coordinate of pixel i, respectively. $I_{max}$, $x_{max}$, and $y_{max}$ denote the maximum intensity, width, and height of the input image. The normalization graph of a cat image from the CIFAR-10 dataset is shown in Figure 3, where the normalized values display a linear relationship with both pixel values and coordinate values.

The hidden layers are empirically configured to comprise 128 nodes, with the potential for subsequent pruning to reduce the parameter count. The output layer is set according to the number of classes, which is 10 for both datasets. The residual activation function $\phi \left(x\right)$ applied at the network edges is defined in Equation (2), as follows:(2)$$\phi \left(x\right)=\omega \left(b\left(x\right)+spline\left(x\right)\right)$$
where *b*(*x*) is the basis function, and *spline*(*x*) is a linear combination of B-splines $B_{i}(x)$, as illustrated in Equations (3) and (4).(3)$$b\left(x\right)=silu\left(x\right)=x/\left(1+e^{-x}\right)$$(4)$$spline\left(x\right)=\sum_{i}c_{i}B_{i}\left(x\right)$$

In Equation (4), $c_{i}$s are trainable. To control the activation function’s overall scale, a weighting factor *ω* is applied. The spline function curve with the order *k* was set to 3 and the grid size was set to 5, shown on the right of Figure 2, as indicated by the dashed line.

To encourage sparsity in the model, we adopt a prediction-based loss function $L_{pred}$ plus *L*1 norm ${\left|\Phi_{l}\right|}_{1}$ of the activation function at a given layer l and entropy regularization $S(\Phi_{l})$ of all KAN layers [16], as presented below in Equation (5):(5)$$L_{total}=L_{pred}+\lambda \left(\mu_{1}\sum_{l=1}^{L-1}{\left|\Phi_{l}\right|}_{1}+\mu_{2}\sum_{l=1}^{L-1}S\left(\Phi_{l}\right)\right)$$
where $L_{pred}$ denotes the categorical cross-entropy loss, as presented in the following Equation (6):(6)$$L_{pred}=-\sum_{i=1}^{N}\sum_{c=1}^{C}y_{i,c}\log(p_{i,c})$$
where $y_{i,c}$ denotes the ground-truth label of the *i*-th sample for class *c*, $p_{i,c}$ represents its corresponding predicted probability, *C* is the number of classes, and *N* is the total number of samples. The entropy regularization $S(\Phi_{l})$ is shown in Equation (7), and L1 norm $|\Phi {|}_{1}$ with ${\;n}_{in}$ inputs and $n_{out}$ outputs is illustrated in Equation (8).(7)$$S\left(\Phi_{l}\right)=-\sum_{i=1}^{n_{in}}\sum_{j=1}^{n_{out}}\frac{\left|\phi_{i,j}\right|}{{\left|\Phi \right|}_{1}}\log(\frac{\left|\phi_{i,j}\right|}{{\left|\Phi \right|}_{1}})$$(8)$${\left|\Phi \right|}_{1}=\sum_{i=1}^{n_{in}}\sum_{j=1}^{n_{out}}{\left|\phi_{i,j}\right|}_{1}$$

The training process of the 2-layer KAN terminates when the accuracy metric plateaus, typically achieving 98% on MNIST and 66% on CIFAR-10 datasets. Then, the optimized model parameters were obtained for computing the contribution score. Considering that the entire image is flattened and directly fed into the KAN as input, the spatial positions of key pixels vary across different images, resulting in inconsistent node assignments in the KAN’s input layer. Consequently, we focus on quantifying the contribution of each pixel in every image to the subsequent layer’s activation, rather than analyzing the contribution of specific nodes in the KAN architecture. As formulated in Equation (1), each input pixel is distributed across two nodes $I_{j,i,x}$ and $I_{j,i,y}$. The contribution score $S_{j,0,i}$ for the *i*-th pixel in the *0*-th layer of the *j*-th image is determined by taking the maximum value between two nodal contribution scores $S_{j,0,i,x}$ and $S_{j,0,i,y}$, as specified in Equation (9). The contribution scores $S_{j,0,i,x}$ and $S_{j,0,i,y}$ represent the maximum values of activation function $\phi_{0,k,i}(\cdot )$ across all nodes k in the subsequent layer, as illustrated in Equations (10) and (11).(9)$$S_{j,0,i}=\max\left(S_{j,0,i,x},S_{j,0,i,y}\right)$$(10)$$S_{j,0,i,x}=\underset{k}{\max}\left(\left|\phi_{0,k,i}\left(I_{j,i,x}\right)\right|\right)$$(11)$$S_{j,0,i,y}=\underset{k}{\max}\left(\left|\phi_{0,k,i}\left(I_{j,i,y}\right)\right|\right)$$

After calculating the contribution score for each pixel, a threshold is set to select key pixels with scores above it. The principle for setting the threshold is to minimize the number of input pixels while ensuring that the classification accuracy is maintained. We empirically evaluate different thresholds by training 2-layer KANs, with results shown in Figure 4 using the MNIST dataset and Figure 5 using the CIFAR-10 dataset. Accuracy sharply declines when the threshold exceeds 0.02, while higher thresholds yield fewer key pixels. To preserve flexibility for the subsequent feature selector, we conservatively chose a sub-critical threshold of 0.01.

### 2.2. EdgeFuseTrim Feature Extractor

The previous section has selected the key pixels that primarily influence the classification results. On the MNIST and CIFAR datasets, the key pixels account for only 14.73% and 13.82% of the total pixels, respectively, significantly reducing input dimensionality. As the key pixels vary across images and are mapped to different input layer nodes, direct pruning of low-contribution nodes is infeasible. Thus, we visualize these feature pixels on the original input images, as shown in the second columns of Figure 6a,b. It is observed that, in the MNIST dataset, these pixels are mostly concentrated along the edges of the digits. In the CIFAR-10 dataset, they are primarily distributed around the object contours and internal details, which also largely correspond to edge-like regions. Therefore, as long as these edge-like pixels can be reliably extracted using a unified method, they can serve directly as input layer nodes.

For computational efficiency, multiple traditional edge detectors were evaluated. Sobel, LoG with zero-crossing, and Canny were selected for visualization due to their superior coverage and low redundancy, as shown in Figure 6. It can be observed that, in both the MNIST and CIFAR datasets, the edges extracted by common detection methods are not identical to the key pixels selected by KAN, but exhibit a high degree of similarity. In this section, we propose Overlap Ratio (OvR) and Redundancy Ratio (RdR) as metrics to evaluate edge detection methods. OvR measures the proportion of pixels shared between the edge detection result Y (yellow in Figure 6) and the key pixels selected by KAN P (purple in Figure 6) relative to the total number of KAN-selected pixels. The formal definition is given in Equation (12). RdR represents the proportion of pixels extracted by the edge detection method that are not covered by the KAN-selected key points relative to the total number of KAN-selected pixels. The formal definition is given in Equation (13).(12)$$OvR=\frac{P\cap Y}{P}$$(13)$$RedR=\frac{Y-P\cap Y}{P}$$

Table 1 presents the average OvR and RdR across labels on the MNIST dataset for different edge detection methods. Sobel and LoG with zero-crossing achieve high OvR but with higher RdR. In contrast, Canny provides lower OvR but maintains lower RdR. The results on the CIFAR dataset are shown in Table 2. As Sobel and LoG with zero-crossing exhibit similar OvR, but Sobel yields lower RdR, only Sobel and Canny are included in Table 2. It can be observed that Sobel achieves better OvR in half of the classes, while Canny performs better in the other half. However, the best OvR from either method rarely exceeds 70%, and both methods exhibit relatively high redundancy. The reason can be intuitively observed from Figure 6: Sobel captures contours well, while Canny better preserves details. However, both methods are sensitive to non-target regions such as sky, grass, and trees, leading to relatively high RdR.

Therefore, we propose a unified feature extractor, EdgeFuseTrim, that integrates Sobel and Canny edge detectors while effectively suppressing non-target edges. The algorithm flow of EdgeFuseTrim is as follows:

Input: Image IOutput: Filtered edge pixels $E_{filtered}$

Initial Edge Detection:

Extract edge pixels using Sobel detector:

(14)$$E_{sobel}\leftarrow Soble(I)$$ Extract edge pixels using Canny detector: (15)$$E_{canny}\leftarrow Canny(I)\;$$

2.Edge Filtering:

Retain only the connected components in $E_{canny}$ that contain at least one pixel coincident with $E_{sobel}$:

(16)$$E_{hybrid}\leftarrow \left\{C\in CC(E_{canny})|C\cap E_{sobel}\neq \emptyset \right\}$$
where CC(·) denotes connected components.

3.Non-target Edge Removal:

Define border regions as outer 16% of image width *w* and height *h*:

(17)$$B\leftarrow \left\{(x,y)|x\;\leq \;0.16w\vee x\;\geq \;0.84w,y\;\leq \;0.16h\vee y\;\geq \;0.84h\right\}$$ Define central region as inner 62% of width and height:(18)$$C\leftarrow \left\{(x,y)|0.19w\leq x\leq 0.19w,0.19h\leq y\leq 0.19h\right\}$$ Remove border-region edges disconnected from C:(19)$$E_{filtered}\leftarrow \left\{e\in E_{hybrid}|e\notin B\;\vee Path(e,C)\;\right\}$$
where Path(e,C) means border edge pixel *e* is connected to *C.* These spatial thresholds, 16% for border and 62% for center, are empirically defined based on the statistical analysis of object locations across the entire CIFAR-10 dataset.

The last column in Figure 6b, generated by the EdgeFuseTrim algorithm, shows that the extracted edges visually resemble those selected by KAN. Table 2 further confirms that this method achieves higher OvR while generally reducing RdR. For the MNIST dataset, this method yields the same result as using Sobel alone, indicating its generality. Although the OvR of EdgeFuseTrim does not reach 100%, the contribution score threshold of 0.01 set in Section 2.1 provides a safety margin exceeding 30% for key pixel preservation. Notably, even the most redundant pixels surrounding feature points retain informational content. Consequently, the minor flaw in OvR is that it has negligible impact on the downstream classification performance.

### 2.3. Input Feature Dimension Unification

Both KAN and MLP networks require fixed input dimensions, whereas the number of feature pixels selected from each image by the EdgeFuseTrim algorithm varies. Therefore, it is necessary to produce a uniform number of the input feature pixels. This involves addressing two issues: first, ensuring that the method for unifying the quantity does not compromise the original feature information; second, selecting an appropriate input dimension $D_{I}$.

The feature pixel information includes both coordinates and pixel values. Directly adding or removing pixels at the original image scale may alter the spatial distribution of the feature pixels. To alter the number of feature pixels while preserving their spatial distribution, bilinear interpolation will be employed to rescale the entire image. Let the number of feature pixels in the *i*-th image be $N_{i}$ and the scaling factor be $S=\sqrt{D_{I}/N_{i}}$. In this way, the feature point mask is also scaled accordingly. However, due to slight discrepancies between the rescaled mask and the target dimension, random removing and zero-padding is applied for adjustment. Since the quantity is minimal, the removal process preserves spatial information integrity, so pixels are randomly selected for deletion, while, to enable gradient computation during training, the padded points are assigned a value of $10^{-7}$ − 1. The overall process is illustrated in Figure 7.

We conducted experiments with varying numbers of input pixels by training 2-layer KANs, and the corresponding accuracy results are shown in Figure 8 and Figure 9. Note that the actual dimension $D_{I}$ should multiply the pixel count by twice the number of channels. On the MNIST dataset, the accuracy stabilized beyond 120 pixels, leading to our final selection of 122 pixels ($D_{I}$ = 122 × 2). On the CIFAR-10 dataset, stable performance was observed above 140 pixels, with 142 pixels ($D_{I}$ = 142 × 6) ultimately chosen.

To evaluate the effectiveness of the method presented in this section, we compared it with directly random removing with zero-padding and peripheral feature point removing with zero-padding methods. We used the PFSKAN (Medium) model structure, as shown in the second diagram of Figure 10, and conducted validation experiments on the MNIST and CIFAR-10 datasets. The experimental results demonstrate that our proposed method achieves superior performance, with validation accuracies of 99.12% and 66.71% on MNIST and CIFAR-10, respectively, outperforming both the random removing (97.06% and 62.53%) and peripheral feature point removing (98.94% and 63.75%) approaches. This demonstrates that the proposed method in this section is more suitable for unifying the input dimensions of the PFSKAN model.

## 3. Experiments

This section evaluates the performance of PFS integrated with KANs and MLPs for the fundamental computer vision task of classification tasks. We conduct comprehensive comparisons with fully connected networks and CNNs in terms of accuracy, parameter efficiency, and training time. To thoroughly assess the generalizability of PFS, experiments are performed not only on standard benchmarks, such as MNIST and CIFAR-10, but also extended to more challenging datasets, including Fashion-MNIST and CIFAR-100, ensuring a robust evaluation across varying datasets.

### 3.1. Model Achitectures

In this study, we designed multiple architectures to evaluate the feature extraction capabilities of PFS against mainstream CNN-based approaches, as well as its compatibility with KANs and classical MLPs. The tested architectures include
PFSKAN: PFS combined with KANs;PFSMLP: PFS integrated with MLPs;Standalone KAN and MLP: one-layer implementations for baseline comparison;ConvMLP: traditional convolutional network with an MLP head;ConvKAN: convolutional network paired with KANs.

Given that MNIST contains low-resolution (28 × 28) single-channel grayscale images, we employed small and medium models, which were subsequently applied to Fashion-MNIST to maintain experimental consistency. In contrast, the CIFAR datasets, while also low-resolution (32 × 32), consist of three-channel RGB color images We therefore introduced large models to adequately process the increased visual complexity. Models of varying scales (small, medium, and large) were uniformly applied to both CIFAR-10 and CIFAR-100 to ensure consistent evaluation conditions.

All models in this work maintain lightweight architectures, with parameters in the million-scale range. The terms “small”, “medium”, and “large” are used in a relative sense within this scale. The detailed model architectures are illustrated in Figure 10. The hyperparameters of all models were tuned through iterative experiments, referencing prior work [25], though further optimization remains possible.

### 3.2. Datasets

The four datasets used in this study are all classic image classification benchmarks. Since the PFS was derived from KAN training on MNIST and CIFAR-10, it is essential to evaluate the PFS performance not only on these source datasets, but also to extend validation to Fashion-MNIST and CIFAR-100 to assess its generalization capability.

Both MNIST and Fashion-MNIST datasets consist of 70,000 grayscale images each, with a standardized split of 60,000 training samples and 10,000 test samples [31,32]. All images share an identical resolution of 28 × 28 pixels and are categorized into 10 classes. While MNIST contains handwritten digits (0–9), Fashion-MNIST comprises apparel categories, resulting in greater visual complexity and marginally higher classification difficulty compared to MNIST.

The CIFAR-10 dataset consists of 60,000 color images, including 50,000 training images and 10,000 test images, divided into 10 categories [33]. Each image is a 32 × 32 RGB representation, exhibiting greater channel depth, and more complex visual content than MNIST.

The CIFAR-100 dataset also contains 60,000 RGB images (50,000 for training and 10,000 for testing) at 32 × 32 resolution, organized hierarchically into 20 superclasses with 5 fine-grained subclasses each [33]. For model consistency (maintaining a 10-class output), we selected the following 10 subclasses: dolphin, sunflower, bottle, orange, table, butterfly, lion, crab, snake, and bicycle, yielding a 6000-image subset (5000 for training and 1000 for test), denoted CIFAR-100_10. All experiments used default datasets splits.

### 3.3. Experimental Setup

All experiments were conducted on a single NVIDIA RTX 4090 GPU (24GB) and 24 CPU cores, using Python 3.9 with Pykan and PyTorch 2.2 as the primary frameworks. The computational environment utilized CUDA 11.8 for GPU acceleration, with auxiliary libraries including OpenCV 4.11, NumPy 1.24, Scikit-learn 1.11, Matplotlib 3.6, and Seaborn 0.13 for image processing, data preprocessing, and visualization. Each model was trained independently under identical computational conditions to ensure fairness.

We employed the AdamW optimizer, with hyperparameters initially set according to the default values suggested in [25]. A grid search was then conducted on the validation set to fine-tune the initial learning rate. As shown in Figure 11, five candidate values (1 × 10^−4^, 5 × 10^−4^, 1 × 10^−3^, 5 × 10^−3^ and 1 × 10^−2^) were evaluated. The results demonstrate that the default setting of 1e-3 yields the best accuracy performance. In addition, the gamma value in an exponential learning rate scheduler was tested using the same grid search strategy over values of 0.75, 0.8, 0.85, and 0.9. As presented in Figure 12, the setting of 0.8 effectively mitigates convergence oscillations, which is consistent with expectations.

In the KAN model, a spline order of three was adopted, which is commonly used and sufficient for fitting smooth curves. The grid size was set to five, as tests with values of five, seven, and nine on the validation set show no significant improvement in accuracy with larger grids, as shown in Table 3, while the number of parameters increased substantially, given that the parameter count of the KAN model is linearly related to the grid size. Hence, a grid size of five was chosen to balance performance and model complexity.

To further reduce the complexity of the KAN model, pruning techniques were applied. The loss function combined cross-entropy with L1 regularization, as defined in Equation (5).

### 3.4. Results

The experiments evaluated the performance of the proposed method and baseline models on the MNIST, Fashion-MNIST, CIFAR-10, and CIFAR-100_10 datasets using multiple metrics: accuracy, precision, recall, and F1 score, as well as parameter count, training time, and evaluation time.

For the MNIST and Fashion-MNIST datasets, where small and medium models were employed, the results of the classification metrics are summarized in Table 4. It can be observed that PFSKAN (Medium) achieved the highest performance across all four metrics on the MNIST dataset, reaching 99.12%. On the Fashion-MNIST dataset, PFSKAN (Medium) obtained a 90.10% accuracy, slightly lower than ConvMLP (Medium)’s 90.12%, but outperformed it in the other three metrics (all 90.10% vs. 90.09%). Notably, ConvMLP (Medium) required 2.58 times more parameters than PFSKAN (Medium).

To facilitate a more intuitive comparison of accuracy across models with similar parameter scales, Figure 13 presents the results using a color-coding scheme where each method is represented by a specific color hue, with lighter shades indicating smaller model sizes. The results demonstrate that PFSKAN models (blue) consistently outperformed comparable PFSMLP (cyan), ConvMLP (yellow), and ConvKAN (magenta) models, with the only exception being PFSKAN (Medium) (dark blue), showing slightly lower performance than ConvMLP (Medium) (dark yellow) on Fashion-MNIST.

The parameter efficiency analysis, shown in Figure 14, reveals that PFSKAN (Medium) not only used fewer parameters but also achieved higher accuracy than a one-layer KAN. The PFSKAN (Small) model required only one-third of the parameters of a one-layer KAN while maintaining comparable accuracy (within 0.1%). Although PFSMLP (Small) showed slightly lower accuracy than a one-layer MLP, it used less than one-sixth of the parameters. Furthermore, PFSKANs consistently demonstrated both smaller parameter sizes and higher accuracy compared to ConvMLPs and ConvKANs of equivalent scales, indicating that PFS can effectively extract features and significantly reduce input dimensionality.

Moreover, PFS appears to be better suited to KAN than to MLP. This may be attributed to two factors: (1) PFS features are selected by KAN and (2) KAN’s higher nonlinearity makes it better equipped to handle pixel-level features. Even though PFS showed slightly less compatibility with MLP, it still maintained high parameter efficiency. For instance, while ConvMLP (Medium) achieved 1.09% and 2.27% higher accuracy than PFSMLP (Medium) on MNIST and Fashion-MNIST, respectively, it required 4.81 times more parameters. Training and evaluation times for all models are shown in Table 5, where PFSKANs consistently consume less time than ConvKANs and ConvMLPs of comparable model size.

Overall, PFS demonstrated significantly higher parameter efficiency than convolutional methods on grayscale images, benefiting from its superior dimensionality reduction capability. The consistent performance patterns between MNIST and Fashion-MNIST further validate the generalizability of PFS.

Our experimental evaluation of the CIFAR-10 and CIFAR-100_10 datasets incorporated large-scale models while maintaining lightweight architectures. As evidenced in Table 6, the proposed PFSKAN (Large) achieved superior performance, with a 78.06% accuracy on CIFAR-10 and 58.92% on CIFAR-100_10, despite the inherent challenges of these more complex color image datasets. The results demonstrate a clear performance hierarchy across model scales. While PFSKANs consistently outperformed alternatives on CIFAR-10, as shown on the left of Figure 15, only medium and large PFSKANs maintained this advantage on the more challenging CIFAR-100_10, as shown on the right of Figure 15. This scaling behavior, coupled with the poor performance of the one-layer KAN, underscores the necessity of deeper and larger architectures for effective feature extraction in complex visual domains.

From a parameter efficiency perspective, shown in Figure 16, PFSKAN (Small) achieved comparable accuracy to the one-layer KAN (within 0.8% difference) while using a mere 27.8% of the parameters, demonstrating remarkable architectural efficiency. Meanwhile, PFSKANs maintain higher accuracy with fewer parameters than ConvKANs and ConvMLPs of comparable model size. PFSMLPs show slightly lower accuracy than other models, remaining highly parameter-efficient. This demonstrates that, on moderately complex color image datasets, PFS is more effective than convolutional methods at reducing input dimensionality, and, again, shows better compatibility with KAN.

It is worth noting that the advantages of PFSKANs in terms of both accuracy and parameter efficiency become increasingly pronounced with the model scale, as PFSKAN (Large) shows substantially better parameter accuracy tradeoffs than the ConvMLP and ConvKAN counterparts. The framework’s computational efficiency is equally noteworthy, with PFSKAN (Large) requiring 37–49% shorter training times than equivalently sized convolutional baselines, as shown in Table 7, while maintaining competitive evaluation speeds. This aligns with the need for larger and deeper models with higher parameter capacity to achieve improved accuracy on complex color images.

The overall performance degradation on the CIFAR-100_10 dataset compared to CIFAR-10 is mainly due to the fact that each class in CIFAR-100_10 contains only 1/10 the number of samples. Nonetheless, the relative advantages of PFS remain consistent across datasets, further supporting its generalizability.

These results collectively validate PFS’s superior dimensionality reduction capability and its particular synergy with KAN architectures, while demonstrating consistent generalization across datasets of varying complexity. The maintained performance advantage on CIFAR-100_10 further confirms the robustness of PFSKAN to data scarcity conditions.

For higher-resolution image datasets such as ImageNet, the key pixels with the highest contribution level selected by KAN are still primarily edge points. However, classification accuracy using only these points does not always surpass that of convolutional networks. When the contribution threshold is lowered, the second-level key pixels include some internal object pixels and a small number of external background pixels. The interpretability and feature extraction methods for these pixels remain under further investigation. Hence, the proposed PFSKAN method, which relies mainly on edge features, is only effective for low-resolution images. To achieve higher accuracy compared to more complex convolutional architectures, such as GoogleNet and ResNet, a hierarchical key pixel extraction strategy must be further refined.

## 4. Conclusions

This work proposes a novel and efficient pixel-level feature selection method based on the interpretability of KAN. Through comparative experiments on the MNIST, Fashion-MNIST, CIFAR-10, and CIFAR-100_10 datasets, we demonstrate that PFS effectively identifies the pixel-level features through EdgeFuseTrim detection due to its interpretability, eliminating the need for time-consuming neural network-based feature extraction and significantly reducing the input dimensionality of subsequent networks for both simple grayscale images and moderately complex color images. When combined with KAN, PFSKAN consistently outperforms convolution-based networks in terms of classification accuracy, parameter efficiency, and training time, with advantages becoming more pronounced as model complexity increases. The superior performance stems from PFS’s fundamental design: it selects comprehensive pixel-level features through a single-pass, computationally efficient edge detection algorithm built upon the interpretability framework of KAN. This contrasts with convolutional approaches that require multiple layers to gradually capture global features through local operations. Our method provides a new interpretable pathway for feature extraction in vision tasks, with experimental validation confirming PFSKAN’s effectiveness for lightweight models and low-resolution image classification. While this work establishes PFSKAN’s efficacy in constrained scenarios, extending and optimizing the hierarchical strategy for more complex vision tasks through architectural enhancements will be the focus of future research. The current results lay important groundwork for developing interpretable, efficient alternatives to convolutional feature extraction paradigms.

## Figures and Tables

**Figure 1 sensors-25-04982-f001:**
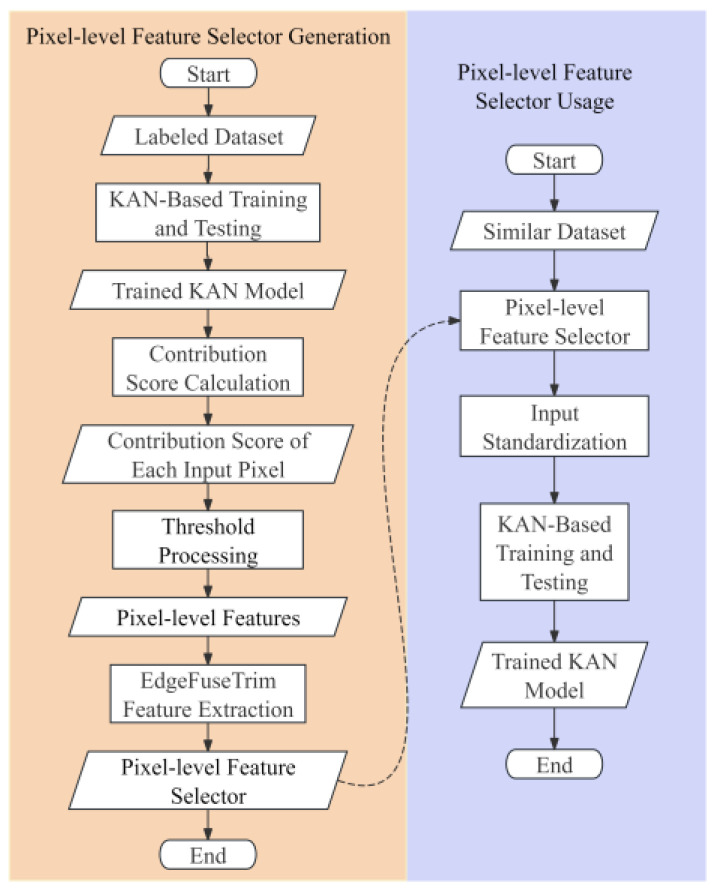
Overall flowchart of PFSKANs. The left yellow region illustrates the generation process of PFS, while the right purple region demonstrates the integrated workflow combining PFS with KANs.

**Figure 2 sensors-25-04982-f002:**
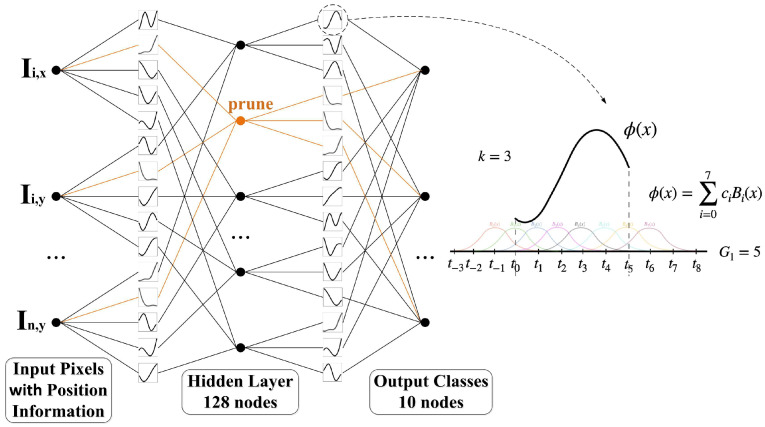
The architecture of KAN model for key pixel selection. The input dimension is 28 × 28 × 2 for MNIST dataset and 32 × 32 × 6 for CIFAR-10 dataset; both use 128 nodes in the hidden layer and 10 output classes. Orange indicates nodes eligible for pruning with negligible contributions.

**Figure 3 sensors-25-04982-f003:**
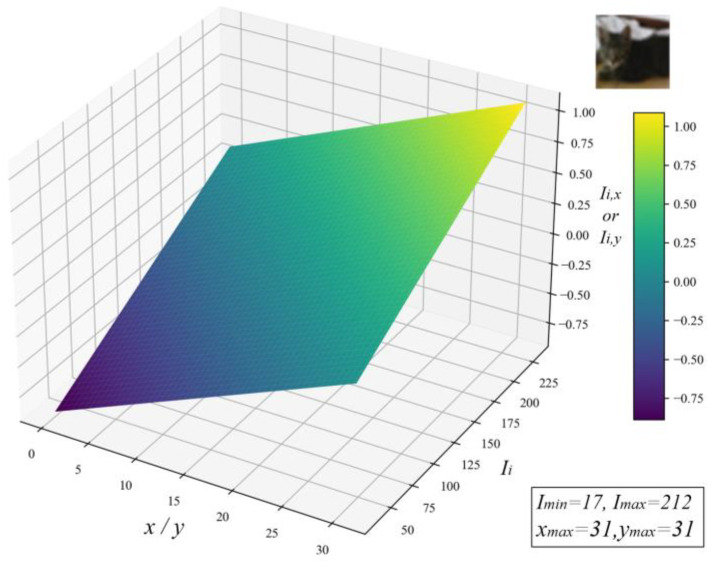
Normalization graph of input pixel and coordinate values.

**Figure 4 sensors-25-04982-f004:**
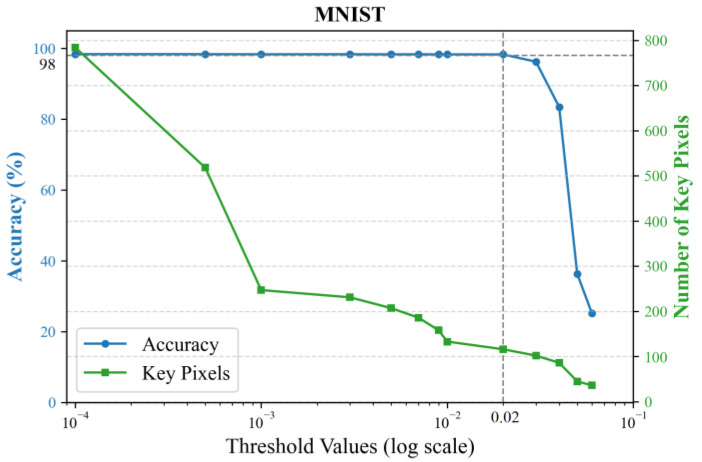
Classification accuracy and the number of key pixels under different thresholds on the MNIST dataset.

**Figure 5 sensors-25-04982-f005:**
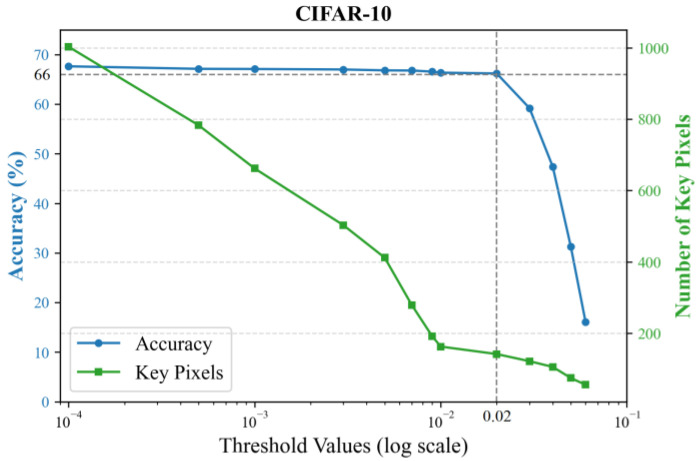
Classification accuracy and the number of key pixels under different thresholds on the CIFAR-10 dataset.

**Figure 6 sensors-25-04982-f006:**
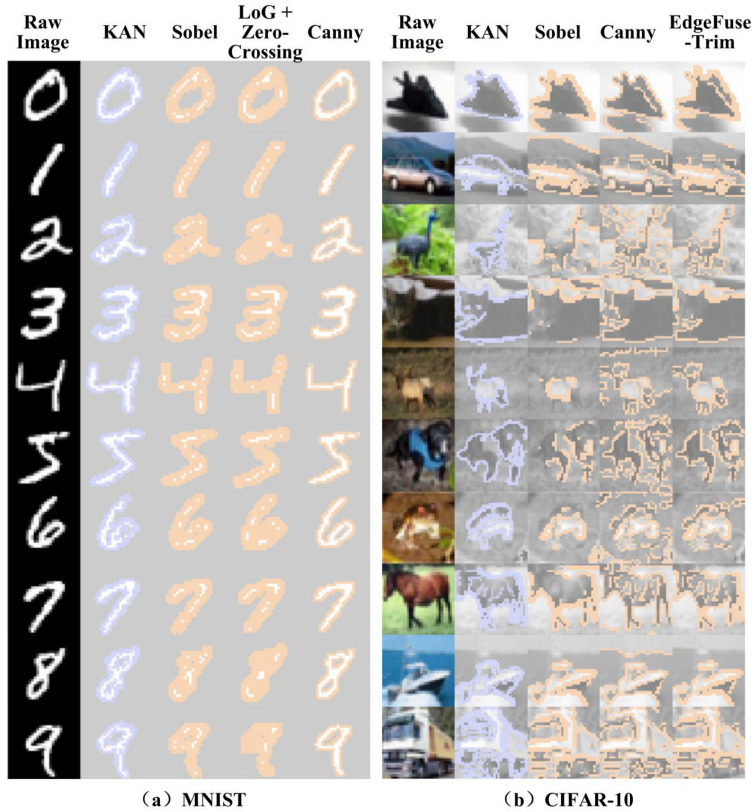
Key pixels selected by KAN mapped onto the original image and comparison with various edge extraction methods: (**a**) on the MNIST dataset and (**b**) on the CIFAR-10 dataset.

**Figure 7 sensors-25-04982-f007:**
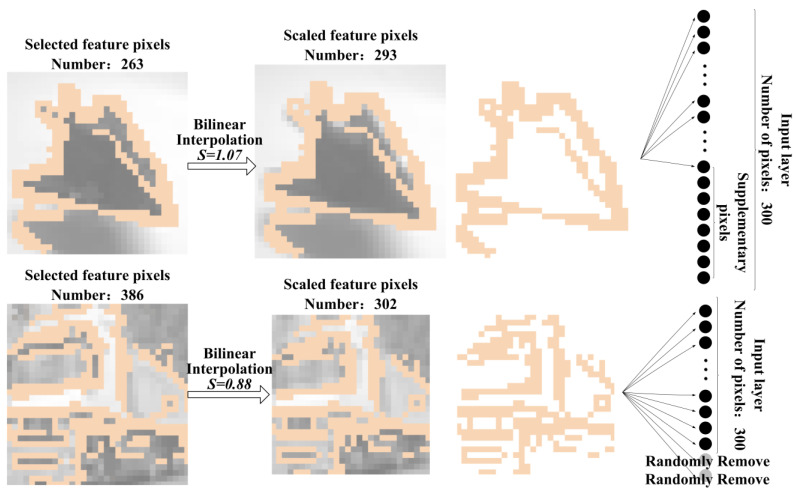
The overall process of input feature dimension unification.

**Figure 8 sensors-25-04982-f008:**
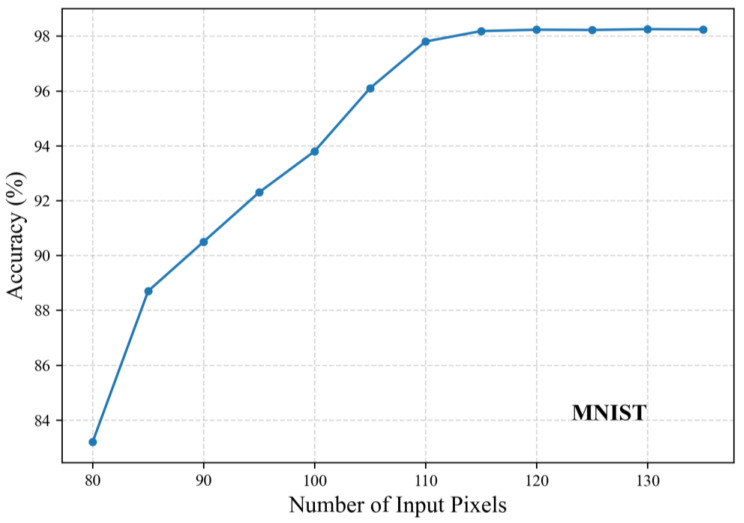
Classification accuracy under different input pixel counts on the MNIST dataset.

**Figure 9 sensors-25-04982-f009:**
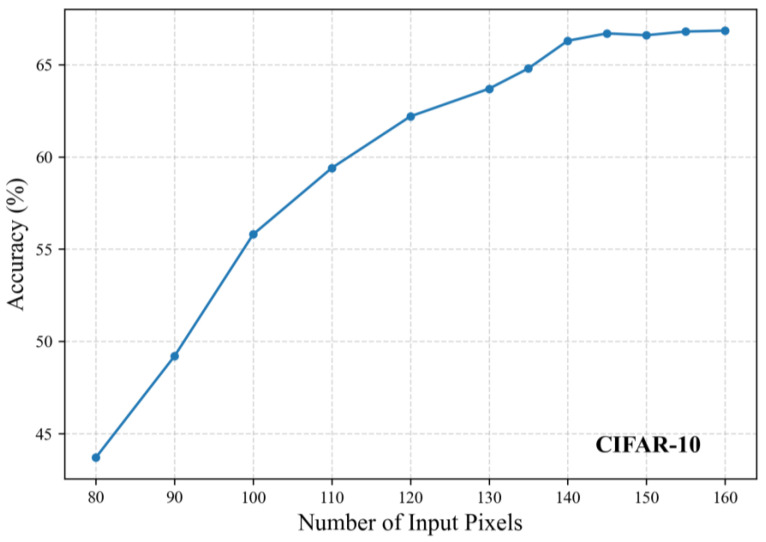
Classification accuracy under different input pixel counts on the CIFAR-10 dataset.

**Figure 10 sensors-25-04982-f010:**
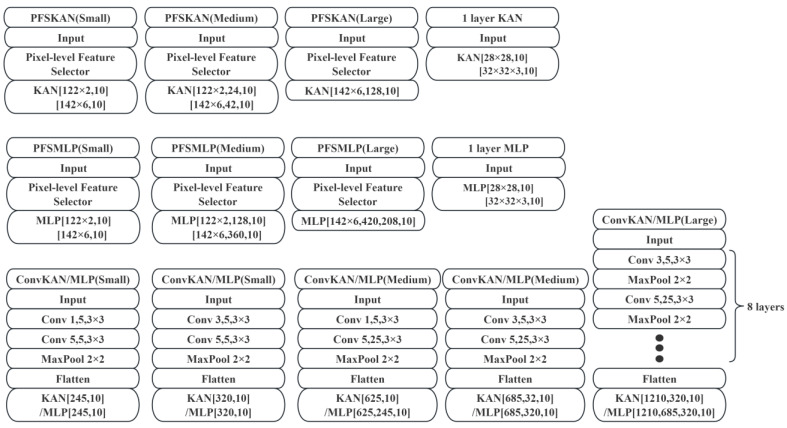
Architectures of the experimental models.

**Figure 11 sensors-25-04982-f011:**
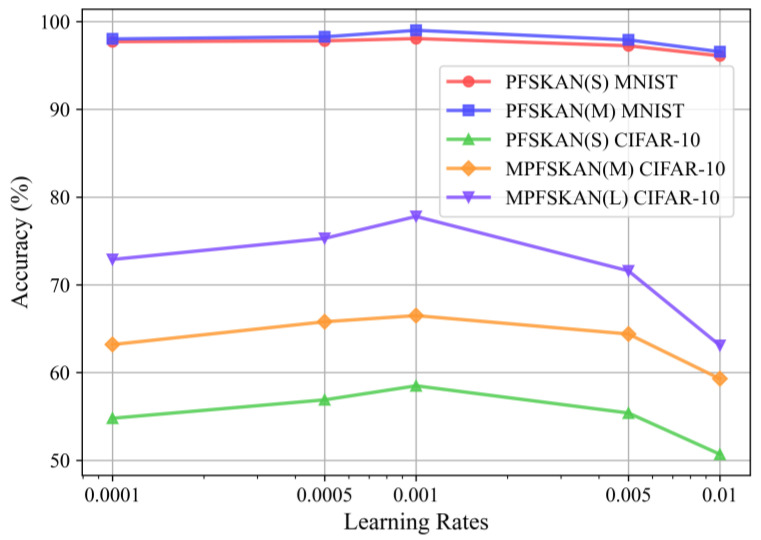
The classification accuracy of different models under various learning rates.

**Figure 12 sensors-25-04982-f012:**
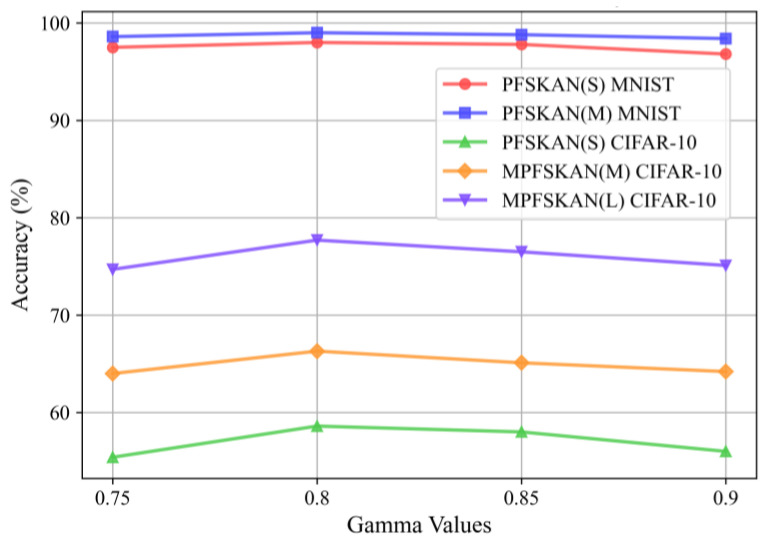
The classification accuracy of different models under various gamma values.

**Figure 13 sensors-25-04982-f013:**
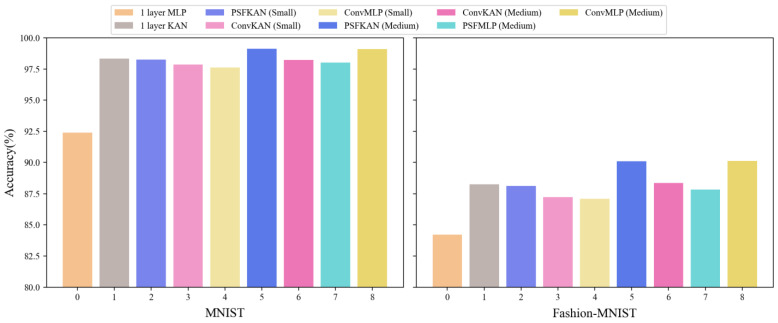
Classification accuracy of different models on the MNIST (**left**) and Fashion-MNIST (**right**) datasets.

**Figure 14 sensors-25-04982-f014:**
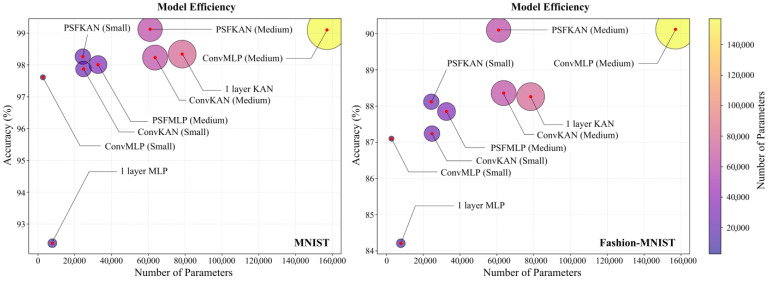
Parameter efficiency and classification accuracy of different models on the MNIST (**left**) and Fashion-MNIST (**right**) datasets.

**Figure 15 sensors-25-04982-f015:**
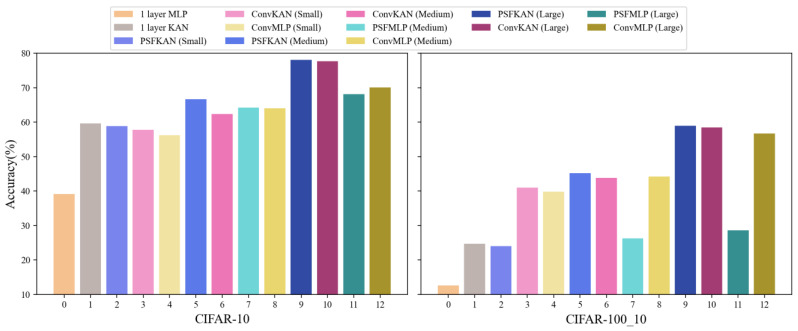
Classification accuracy of different models on the CIFAR-10 (**left**) and CIFAR-100_10 (**right**) datasets.

**Figure 16 sensors-25-04982-f016:**
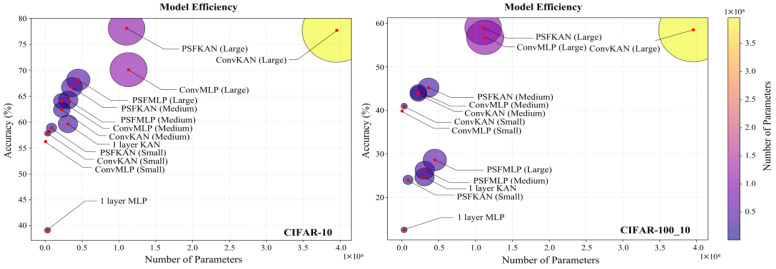
Parameter efficiency and classification accuracy of different models on the CIFAR-10 (**left**) and CIFAR-100_10 (**right**) datasets.

**Table 1 sensors-25-04982-t001:** Evaluation of edge detection methods on the MNIST datasets.

Label	Sobel		LoG + Zero-Crossing		Canny	
	OvR	RdR	OvR	RdR	OvR	RdR
0	0.9917	0.8833	0.9902	1.0417	0.5583	0.3333
1	1	0.5316	0.9873	0.7215	0.557	0.2025
2	0.9774	0.6917	0.985	0.9398	0.5113	0.2556
3	0.9868	0.5132	0.9605	0.6447	0.4539	0.1842
4	0.9677	0.9785	0.9785	1.3011	0.3978	0.5054
5	1	0.6667	1	0.8968	0.5238	0.2462
6	0.9441	0.7447	0.9716	0.8964	0.4242	0.3367
7	0.9808	0.8269	0.9615	0.9423	0.5577	0.2885
8	0.9421	0.5868	0.9835	0.8182	0.4298	0.2975
9	0.9608	0.7745	0.9618	0.9902	0.4706	0.3235

**Table 2 sensors-25-04982-t002:** Evaluation of edge detection methods on the CIFAR-10 datasets.

Label	Sobel		Canny		EdgeFuseTrim	
	OvR	RdR	OvR	RdR	OvR	RdR
airplane	0.7302	0.9841	0.3571	0.8254	0.8413	1.0571
automobile	0.6726	1.6903	0.4602	1.3363	0.7685	1.1621
bird	0.4505	1.1443	0.6021	2.4134	0.7433	1.1124
cat	0.4479	1.4375	0.6256	1.3958	0.7671	1.1288
deer	0.4046	0.5846	0.4615	2.1923	0.7538	0.8615
dog	0.4287	1.1517	0.5238	2.3927	0.7139	1.2521
frog	0.4151	0.8403	0.4202	1.8168	0.7042	1.0308
horse	0.6869	0.6822	0.4112	0.8972	0.8162	0.7132
ship	0.6803	0.7143	0.3741	0.8776	0.7871	0.7247
truck	0.5721	0.6006	0.4172	0.6124	0.7793	0.6627

**Table 3 sensors-25-04982-t003:** The classification accuracy of different models under various grid sizes.

Datasets	Model	Accuracy (%)		
		5	7	9
MNIST	PFSKAN(S)	98.21	98.23	98.18
	PFSKAN(M)	99.08	99.10	99.11
CIFAT-10	PFSKAN (S)	58.85	58.86	58.82
	PFSKAN (M)	66.72	66.71	66.72
	PFSKAN (L)	78.03	78.05	78.06

**Table 4 sensors-25-04982-t004:** Comparison with different models on the MNIST and Fashion-MNIST datasets.

Model	MNIST (%)				Fashion-MNIST (%)				Parameters
	Accuracy	Precision	Recall	F1 Score	Accuracy	Precision	Recall	F1 Score	
1-Layer MLP	92.40	92.32	92.32	92.32	84.21	84.20	84.20	84.20	7850
1-Layer KAN	98.34	98.34	98.34	98.34	88.26	88.24	88.23	88.23	78,400
PFSMLP (S)	88.26	88.26	88.26	88.26	78.52	78.52	78.52	78.52	1230
PFSMLP (M)	98.01	98.00	98.00	98.00	87.85	87.85	87.85	87.85	32,650
**PFSKAN(S)**	**98.26**	**98.26**	**98.26**	**98.26**	**88.12**	**88.11**	**88.11**	**88.11**	**24,400**
**PFSKAN(M)**	**99.12**	**99.12**	**99.12**	**99.12**	**90.10**	**90.10**	**90.10**	**90.10**	**60,960**
ConvKAN (S)	97.87	97.87	97.87	97.87	87.24	87.21	87.22	87.21	24,780
ConvKAN (M)	98.23	98.23	98.23	98.23	88.36	88.36	88.36	88.36	63,700
ConvMLP (S)	97.61	97.61	97.61	97.61	87.10	87.03	87.03	87.03	2740
ConvMLP (M)	99.10	99.10	99.10	99.10	90.12	90.09	90.09	90.09	157,030

**Table 5 sensors-25-04982-t005:** Comparison of time and epochs with different models on the MNIST and Fashion-MNIST datasets.

Model	MNIST			Fashion-MNIST			Parameters
	Epochs	Training Time	Evaluation Time	Epochs	Training Time	Evaluation Time	
1-Layer MLP	10	2.58 min	2.21 s	10	2.69 min	2.14 s	7850
1-Layer KAN	10	5.65 min	4.52 s	10	6.02 min	4.76 s	78,400
PFSMLP (S)	10	0.87 min	1.79 s	10	1.03 min	1.85 s	1230
PFSMLP (M)	50	8.31 min	3.24 s	50	9.13 min	3.32 s	32,650
**PFSKAN(S)**	**10**	**2.56 min**	**3.16 s**	**10**	**2.96 min**	**3.19 s**	**24,400**
**PFSKAN(M)**	**50**	**12.42 min**	**4.86 s**	**50**	**12.42 min**	**4.96 s**	**60,960**
ConvKAN (S)	10	4.27 min	3.19 s	10	4.64 min	3.27 s	24,780
ConvKAN (M)	50	17.25 min	4.93 s	50	17.81 min	4.98 s	63,700
ConvMLP (S)	10	3.12 min	3.87 s	10	3.74 min	4.01 s	2740
ConvMLP (M)	50	18.34 min	5.06 s	50	19.07 min	5.25 s	157,030

**Table 6 sensors-25-04982-t006:** Comparison with different models on the CIFAR-10 and CIFAR-100_10 datasets.

Model	CIFAR-10 (%)				CIFAR-100_10 (%)				Parameters
	Accuracy	Precision	Recall	F1 Score	Accuracy	Precision	Recall	F1 Score	
1-Layer MLP	39.12	39.51	39.43	39.47	12.57	12.57	12.57	12.57	30,730
1-Layer KAN	59.65	59.57	59.39	59.48	24.68	24.68	24.67	24.67	307,200
PFSMLP (S)	32.46	32.46	32.46	32.46	12.64	12.53	12.53	12.53	8530
PFSMLP (M)	64.25	64.23	64.22	64.22	26.27	26.27	26.27	26.27	310,690
PFSMLP (L)	68.14	68.14	68.14	68.14	28.61	28.61	28.61	28.61	447,918
**PFSKAN (S)**	**58.87**	**58.86**	**58.86**	**58.86**	**24.04**	**24.01**	**24.01**	**24.01**	**85,200**
**PFSKAN (M)**	**66.71**	**66.70**	**66.70**	**66.70**	**45.21**	**45.17**	**45.15**	**45.16**	**362,040**
**PFSKAN (L)**	**78.06**	**78.04**	**78.04**	**78.04**	**58.92**	**58.87**	**58.83**	**58.85**	**1,103,360**
ConvKAN (S)	57.82	57.82	57.82	57.82	40.96	40.91	40.90	40.90	32,390
ConvKAN (M)	62.36	62.32	62.32	62.32	43.78	43.78	43.78	43.78	223,690
ConvKAN (L)	77.68	77.65	77.66	77.65	58.47	58.43	58.43	58.43	3,956,020
ConvMLP (S)	56.24	56.24	56.24	56.24	39.83	39.74	39.75	39.74	3580
ConvMLP (M)	64.03	64.07	64.06	64.06	44.17	44.15	44.15	44.15	224,020
ConvMLP (L)	70.10	70.10	70.10	70.10	56.73	56.73	56.73	56.73	1,129,810

**Table 7 sensors-25-04982-t007:** Comparison of time and epochs with different models on the CIFAR-10 and CIFAR-100_10 datasets.

Model	CIFAR-10			CIFAR-100_10			Parameters
	Epochs	Training Time	Evaluation Time	Epochs	Training Time	Evaluation Time	
1-Layer MLP	10	4.78 min	3.10 s	50	4.46 min	3.01 s	30,730
1-Layer KAN	10	11.25 min	5.79 s	50	11.12 min	5.20 s	307,200
PFSMLP (S)	10	1.21 min	2.04 s	50	1.21 min	1.92 s	8530
PFSMLP (M)	50	17.31 min	6.01 s	250	16.05 min	5.59 s	310,690
PFSMLP (L)	150	39.39 min	8.43 s	750	36.26 min	8.14 s	447,918
**PFSKAN (S)**	**10**	**4.12 min**	**4.67 s**	**50**	**3.68 min**	**4.16 s**	**85,200**
**PFSKAN (M)**	**50**	**18.73 min**	**6.35 s**	**250**	**16.32 min**	**5.89 s**	**362,040**
**PFSKAN (L)**	**150**	**52.06 min**	**10.39 s**	**750**	**51.57 min**	**10.10 s**	**1,103,360**
ConvKAN (S)	10	7.31 min	4.06 s	50	7.06 min	3.91 s	32,390
ConvKAN (M)	50	26.16 min	6.13 s	250	27.64 min	4.83 s	223,690
ConvKAN (L)	150	104.24 min	12.36 s	750	98.79 min	12.41 s	3,956,020
ConvMLP (S)	10	5.41 min	4.38 s	50	5.12 min	4.12 s	3580
ConvMLP (M)	50	21.25 min	6.97 s	250	20.65 min	6.35 s	224,020
ConvMLP (L)	150	85.34 min	11.15 s	750	82.97 min	11.07 s	1,129,810

## Data Availability

The original data presented in the study are openly available in [The MNIST Database of Handwritten Digits] at [http://yann.lecun.com/exdb/mnist/], [Fashion-mnist: A novel image dataset for benchmarking machine learning algorithms] at [10.48550/arXiv.1708.07747] or [http://fashion-mnist.s3-website.eu-central-1.amazonaws.com/] or [https://github.com/zalandoresearch/fashion-mnist], and [Learning Multiple Layers of Features from Tiny Images] at [https://www.cs.toronto.edu/~kriz/learning-features-2009-TR.pdf] or [https://www.cs.toronto.edu/~kriz/cifar-10-python.tar.gz] and [https://www.cs.toronto.edu/~kriz/cifar-100-python.tar.gz].

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
