# Peer review of "PFSKANs: A Novel Pixel-Level Feature Selection Model Based on Kolmogorov–Arnold Networks"

_sensors, 2025, doi:10.3390/s25164982_

Round 1

Reviewer 1 Report

Comments and Suggestions for Authors

KAN is interesting as a new promising method alternative to CNN. But, it requires more vigorous research to prove its ability over the widely accepted CNN model. I found in Figure 5 authors are trying o compare their model with other edge extraction method. But the initial goal of this paper is to prove the ability of KAN over CNN.

So, I recommend you train your data with some other CNN based model such as DeepLabv3 (or other models) and then compare your result with KAN.

I also found that there are a lot of noises in the edge detection method. It's failed to distinguish the edge or too thick of the edge means that it considers the background also. Authors didn’t mention about their limitation and future goal.

The paper also lacks data preparation strategy. It is difficult to understand the uniqueness of the KAN model. The image quality needs to improve.

Author Response

Comments 1: KAN is interesting as a new promising method alternative to CNN. But, it requires more vigorous research to prove its ability over the widely accepted CNN model. I found in Figure 5 authors are trying o compare their model with other edge extraction method. But the initial goal of this paper is to prove the ability of KAN over CNN.

Response 1: Thank you for pointing this out. We agree that it requires more vigorous research to prove its ability over the widely accepted CNN model. In this paper, we have demonstrated that our method, PFSKAN, only outperforms CNN in terms of classification accuracy and computational efficiency just in low-resolution image classification tasks. This suggests a potential alternative approach to CNNs. However, for higher resolution image datasets such as ImageNet, our method does not always outperform CNN, and further improvements are needed to enhance its performance. This limitation is discussed in the last paragraph of Section 3.4 (page 18, line 487-496) and in line 524-527 of Section 4 (page 20).

In Section 2.1 (page 4), we found through the interpretability of KAN that the edge-like pixels in the image contribute the most to the KAN network as key pixels. Therefore, in Section 2.2 (page 8), where Figure 5 (page 8, line 218) is presented, our goal is to propose a method that can accurately select key pixels. This allows the input to the other KAN network to shift from all pixels to just the key pixels, significantly reducing the input dimension. Thus, Figure 5 compares our proposed EdgeFuseTrim edge detection method with other edge detection methods, demonstrating that the points detected by our method most closely align with the key pixels.

The comparison experiment between the entire PFSKAN and CNNs is presented in Table 4-7 and Feature 13-16 in Section 3.4 (page 16-20).

Comments 2: So, I recommend you train your data with some other CNN based model such as DeepLabv3 (or other models) and then compare your result with KAN.

Response 2: Thank you for providing the models such as DeepLabv3. Our method was first validated on low-resolution datasets, MNIST (28×28 Grayscale) and CIFAR (32×32 RGB). The CNN models used for the comparative experiments are relatively small, as shown in the third row of Figure 10 (page 13). DeepLabv3 will be used for comparison in the next research phase with higher resolution image datasets.

Comments 3: I also found that there are a lot of noises in the edge detection method. It's failed to distinguish the edge or too thick of the edge means that it considers the background also. Authors didn’t mention about their limitation and future goal.

Response 3: We agree that it considers the background also. The key pixels selected based on the interpretability of KAN not only include the edges but also the points surrounding the edges. This indicates that it is not just the precise edges, but also the comparison of pixel values between the edges and their surrounding points that contribute key information to KAN. The edges extracted by the Sobel method tend to be thicker, while our EdgeFuseTrim method also integrates the Sobel method. Our goal is to align with the key pixels, rather than strictly selecting only the edges. To be more accurate in description, we changed "edge" to "edge-like" in the first paragraph of Section 2.2 (page 8, line 215-216).

Thank you for pointing the limitation and future goal. The limitations of our method do exist. For higher-resolution image datasets, such as ImageNet, our method is not always superior to convolutional approaches. We have included an analysis of these limitations and future goal in the last paragraph of Section 3.4 (page 18,line 487-496), and in line 524-527 of Section 4 (page 20).

Comments 4:The paper also lacks data preparation strategy. It is difficult to understand the uniqueness of the KAN model. The image quality needs to improve.

Response 4: Thank you for pointing this. This paper is the first attempt at this kind of method, with the goal of initially validating its feasibility on low-resolution image datasets. As such, it covers nearly all commonly used low-resolution datasets, including MNIST, Fashion-MNIST, CIFAR-10, and CIFAR-100. We have validated the capabilities of PFSKAN on low-resolution images, laying the foundation for further method improvements on higher resolution image datasets. Our future work will expand to more general datasets.

If the "image" you mentioned refers to the figures, we have also revised all the features except for Figure 1 to improve their visibility and quality.

Reviewer 2 Report

Comments and Suggestions for Authors

Title "PFSKANs: A Novel Pixel-level Feature Selection Model Based on Kolmogorov-Arnold Networks" submit to Sensor MDPI ensure a quality of feature selection method for KAN which provide an alternative for multilevel perceptron. The promised the a pixel level feature selection method which shows better result in Comparison to CNN over low resolution images.  The points the needs to address to get remark to the point for acceptance: 

1) While selecting features based modified KAN interpretability technique, how computational cost varies for block level of pixel values. Mention the pixel characterstic are used to make the procedure optimal. 

2) Unifying the varying pixels values in the input, if the authors wish to respond its feature normalization. Show its graphical representation. How the loss is calculated in case any particular image have higher pixel values vs image having lower pixel values. 

3) As per the flow model in fig 1, the authors should have always label data? How the result and performance can tested on unlabeled data?  Fig.2 also required to redraw again as printed the text light color text will not be visible. 

4) The authors also ensured to proposed a method of unifying the input dimension but how does this methods is validity. does it provide the comparative analysis with existing methods. 

The authors need to provide comparative prospectus and abalation studies. 

Comments on the Quality of English Language

English is fine. 

Author Response

Comments 1: While selecting features based modified KAN interpretability technique, how computational cost varies for block level of pixel values. Mention the pixel characterstic are used to make the procedure optimal.

Response 1: We considered “block level of pixel values” you referred to may probably means a region contains the central pixel and its surrounding neighborhood or a region similar to the kernel in CNNs, then the modified KAN's interpretability technique does not involve the concept of blocks. The interpretability feature selection of KAN is done by calculating the contribution value of each input pixel alone, selecting pixels with high contributions. The calculation of each input pixel's contribution value may be difficult to understand, but it can be referenced in the KAN structure diagram shown in Figure 2 (page 5, line 139). The contribution value is computed by the activation function , which is only related to a single input pixel itself and does not involve the computation of a block, with the specific computation process described in the 5th paragraph of Section 2.1 (page 6-7, line 178-191). If block level of pixel values refers to the precision of pixel values normalized to the [-1, 1] range, then, by convention, all values are represented using 32-bit floating-point numbers.

We have re-described the benefits of pixel characterstic in the Section 4 (page 20, line 510-514,518-521). This is also described in the fourth paragraph of the introduction in Chapter 1(page 2, line 86-94). Through the interpretability technique of the modified KAN, we found that the feature pixels extracted by the neural network strongly overlap with the edge pixels. This not only provides interpretability for the feature extraction of the black-box neural network, but also, when using KAN for image classification in the future, allows us to input only the edge-like pixels rather than all pixels. This reduces the network's dimensionality and significantly speeds up the computation. We hope we have provided a clear answer to your comment.

Comments 2: Unifying the varying pixels values in the input, if the authors wish to respond its feature normalization. Show its graphical representation. How the loss is calculated in case any particular image have higher pixel values vs image having lower pixel values.

Response 2: Thank you for pointing this out. Normalizing pixel values to the range [-1, 1] is because the activation function of the KAN includes a B-spline function, which has an input range of [-1, 1]. We have added Figure 3, the Normalization graph, after the second paragraph of Section 2.1 (page 5, line 158), which more intuitively demonstrates that we use linear normalization with pixel values and spatial location information.

If the loss you referred to is the loss function used for training the classification network, as shown in Equations (5) and (6) in the fourth paragraph of Section 2.1 (page 6, line 171, 172), primarily consists of categorical cross-entropy loss , which is insensitive to the magnitude of pixel values. If it refers to the normalization loss, then Min-Max Scaling can shift the pixel histogram to the central region, and the neural network has adaptability to the image brightness.

Comments 3: As per the flow model in fig 1, the authors should have always label data? How the result and performance can tested on unlabeled data?  Fig.2 also required to redraw again as printed the text light color text will not be visible.

Response 3: The method presented in this paper is based on supervised learning, so the training datasets are labeled datasets. The trained model can then classify unlabeled images. To test the classification accuracy, the test datasets have its labels hidden. Then the trained model predicts the classification results for the test sets. These predicted results are compared with the true labels to calculate accuracy. The results in Table 4 (page 16, line 443) and Table 6 (page 19, line 500) in Section 3.4, are calculated in this way.

Thank you for pointing out the issue with Figure 2. We have revised Figure 2 located after the first paragraph of Section 2.1 (page 5, line 139).

Comments 4: The authors also ensured to proposed a method of unifying the input dimension but how does this methods is validity. does it provide the comparative analysis with existing methods. The authors need to provide comparative prospectus and abalation studies.

Response 4: Thank you for pointing this out. The methods for unifying input dimensions can be broadly categorized into two classes: the first based on image resizing through scaling, and the second involving padding followed by random or region-specific feature reduction. In this paper, the input consists of selected edge feature points. The second category methods, when random removing feature points directly, may damage the spatial properties of the whole feature points. Using border removing may also result in the loss of important information, as some crucial details may be located at the edges of the image. Therefore, the method proposed in this paper is based on the first category, utilizing bilinear interpolation to scale the image. We have added experimental validation in the last paragraph of Section 2.3 (page 11,line 304-313), with the following description:

To evaluate the effectiveness of the method presented in this section, we compared it with directly random removing with zero-padding and peripheral feature point removing with zero-padding methods. We used the PFSKAN (Medium) model structure, as shown in the second diagram of Figure 10, and conducted validation experiments on the MNIST and CIFAR-10 datasets. The experimental results demonstrate that our proposed method achieves superior performance, with validation accuracies of 99.12% and 66.71% on MNIST and CIFAR-10 respectively, outperforming both random removing (97.06%, 62.53%) and peripheral feature point removing (98.94%, 63.75%) approaches. This demonstrates that the proposed method in this section is more suitable for unifying the input dimensions of the PFSKAN model.

The comparative prospectus and abalation studies of proposed method PFSKAN are presented in Section 3 (page 13-20).

Reviewer 3 Report

Comments and Suggestions for Authors

The manuscript introduces PFSKANs, a novel model for pixel-level feature selection in image classification tasks, leveraging the interpretability of Kolmogorov-Arnold Networks (KANs). The authors propose a different approach from conventional CNNs and Transformers, aiming to identify key pixels directly at the input level through a mathematically grounded and interpretable mechanism.

Strengths:

  1. The paper presents a compelling alternative to CNNs and Transformers by utilizing KANs for feature selection. This direction is timely and relevant, especially given the growing interest in interpretable and efficient models in computer vision.
  2. The proposed method is well-explained, with a thorough description of the pixel selection process, the simplification techniques adapted from KANs, and the rationale behind the experimental setup. The choice of datasets is appropriate and allows for a fair comparison across different image complexities.
  3. The experiments demonstrate that PFSKANs achieve performance comparable to CNNs in terms of accuracy, parameter efficiency, and training time. The results are discussed in a balanced and transparent manner.
  4. The article aligns well with the aims and scope of Sensors, particularly the Sensing and Imagingsection and the special issue on image feature extraction.

Critical remarks:

  1. The introduction would benefit from stronger and more up-to-date literature support. For example, the important claim ending on line 43 is backed by a single conference paper from 2014, and the statement ending on line 61 lacks any citation. Incorporating recent and high-impact references would strengthen motivation and context.
  2. The visual presentation of the paper could be improved. For instance, Figure 2appears hastily prepared and lacks the polish expected in a peer-reviewed journal.
  3. The use of the term “higher resolution” when comparing 28×28 and 32×32 images is somewhat misleading. The key distinction lies in the color channels (grayscale vs. RGB)rather than resolution per se.
  4. In subsection 3.3, while the hardware is described, the paper lacks details on the computational environment, such as the software frameworks, libraries, and implementation tools used. Additionally, while the choice of training (hyper)parameters seems reasonable, a brief justification would enhance the credibility of the experimental design.
  5. The paper would benefit from a more explicit discussion of method limitations, especially in the context of resolution. For example, how well does the method generalize to higher-resolution or more complex datasets beyond CIFAR-100?

Author Response

Comments 1: The introduction would benefit from stronger and more up-to-date literature support. For example, the important claim ending on line 43 is backed by a single conference paper from 2014, and the statement ending on line 61 lacks any citation. Incorporating recent and high-impact references would strengthen motivation and context.

Response 1: Thank you for pointing this out. We agree with this comment. Therefore, We have replaced reference [1] in line 43 with the more recent and authoritative work:
Nixon, M.; Aguado, A. Feature Extraction and Image Processing for Computer Vision. Academic Press: 2019.

This book represents one of the most up-to-date comprehensive references in traditional feature extraction methods. While the most active research period for traditional feature extraction methods was before 2020, these methods remain valuable for their computational efficiency and interpretability - characteristics that complement neural network-based approaches. In Section 2.2, we employed edge detection, one of these traditional techniques.

The statement ending on line 61 now cites the reference [11]: "Venkatesh, B.; Anuradha, A. A review of feature selection and its methods. Cybernetics and Information Technologies 2019, 19 (1), 3-26." Thank you for pointing out this detail.

At the same time, we have replaced the original references [5], [6], [9], [10], [12], [13], [14], and [23] with either more high-impact or more recent ones, which have been highlighted in red in the Reference section on pages 21-22.

Comments 2: The visual presentation of the paper could be improved. For instance, Figure 2appears hastily prepared and lacks the polish expected in a peer-reviewed journal.

Response 2: Agree. We have redrawn the Figure 2 on page 5, line 139. The previous curves were unclear, so it has been changed to a straight-line visual style from the original KAN[16] paper for better clarity. The font has also been bolded and enlarged. Additionally, the original image in the lower-left corner has been moved to the right and enlarged, making it easier to view the details. The corresponding descriptions have also been modified in line 147 on page 5 and line 168 on page 6 to match the changes in the Figure 2.

Meanwhile, improvements have also been made to other figures. For Figure 4 and Figure 5 on page 7, the size and font have been adjusted. For Figure 6 (a) and (b) on page 8, the first column, due to the smaller pixel size of the dataset, was kept at the original resolution without interpolation, making it somewhat blurry. However, this was intentional by the authors, as it makes it easier to compare edge extraction in the subsequent columns. Only the font has been bolded and enlarged. For Figure 7 on page 11, the style follows that of Figure 2, with changes made to the input layer and font size. For Figures 8 and 9 on page 12, the size and font have been adjusted. Figure 10 on page 13 has bolded fonts. Figure 11 on page 14 and Figure 12 on page 15 are newly added (to be mainly described in the Response 4). Figure 13 on page 17 has an improved resolution. For Figure 14 on page 18, the font size has been increased. Finally, Figure 15 on page 20 has an improved resolution, and the font size for Figure 16 on page 20 has been increased. All the modified figures have their captions marked in red. Only the caption content of Figure 2 has been modified.

Comments 3: The use of the term “higher resolution” when comparing 28×28 and 32×32 images is somewhat misleading. The key distinction lies in the color channels (grayscale vs. RGB) rather than resolution per se.

Response 3: Thanks for pointing that misleading, we agree that both MNIST and CIFAR contain low-resolution images. In Section 3.1 (paragraph 2, line 335-339), we now emphasize their key difference: MNIST's single-channel grayscale versus CIFAR's three-channel RGB images. Correspondingly, in Section 3.2 (paragraph 3, line 363), we have removed the discussion regarding “higher resolution”.

Comments 4: In subsection 3.3, while the hardware is described, the paper lacks details on the computational environment, such as the software frameworks, libraries, and implementation tools used. Additionally, while the choice of training (hyper)parameters seems reasonable, a brief justification would enhance the credibility of the experimental design.

Response 4: Thank you for pointing this out. We agree with this comment. We have made the following revisions to Section 3.3 (page 14-15, line 373-400): In the first paragraph, we have supplemented the details about the Python version, the deep learning frameworks required (PyTorch 2.2, PyKAN), the corresponding CUDA version, and the necessary open-source libraries. In the second paragraph, we have specified that the two hyperparameters of the AdamW optimizer were initially determined based on [25] and then further refined through experiments using the grid search method, the corresponding experimental results are now presented in Figures 11 and 12. In the third paragraph, we explained that the spline order of 3 in the KAN model is a common practice, sufficient curve-fitting capability. Regarding the grid size, we set it to 5 after testing with values of 7 and 9 in Table 3, where no significant improvement in accuracy was observed, but the number of parameters increased dramatically. Thus, we determined 5 as the optimal value.

Comments 5: The paper would benefit from a more explicit discussion of method limitations, especially in the context of resolution. For example, how well does the method generalize to higher-resolution or more complex datasets beyond CIFAR-100?

Response 5: Thank you for pointing this. The limitations of our method do exist. For higher-resolution image datasets, such as ImageNet, our method is not always superior to convolutional approaches. We have included an analysis of these limitations in the last paragraph of Section 3.4 (page 18, line 487-496), and the full content is as follows:

For higher-resolution image datasets such as ImageNet,  the key pixels with the highest contribution level selected by KAN are still primarily edge points. However, classification accuracy using only these points does not always surpass that of convolutional networks. When the contribution threshold is lowered, the second-level key pixels include some internal object pixels and a small number of external background pixels. The interpretability and feature extraction methods for these pixels remain under further investigation. Hence, The proposed PFSKAN method, which relies mainly on edge features, is only effective for low-resolution images. To achieve higher accuracy compared to more complex convolutional architectures such as GoogleNet and ResNet, a hierarchical key pixel extraction strategy must be further refined.

Reviewer 4 Report

Comments and Suggestions for Authors

The subject of the paper is interesting, but the following aspects must be improved:

-the interpretability part is not clearly explained: how can the user benefits from the interpretability part in order to understand the result of the model?

-compare the obtained results with other existing ones in order to show the benefits of the proposed method

-how can be applied the proposed method for other datasets (on the same task - classification)

Author Response

Comments 1: the interpretability part is not clearly explained: how can the user benefits from the interpretability part in order to understand the result of the model?

Response 1: Thank you for pointing this out. We have re-described the benefits of interpretability in the section 4 (page 20, line 510-514, 517-520). The interpretability of KAN is briefly introduced in the third paragraph of Section 1 (page 2, lines 73-75), while the modified KAN interpretability technique is described in detail in paragraphs 5 and 6 of Section 2.1 (page 6, line 178-199 ). The advantages of interpretable feature points are discussed in the fourth paragraph of Section 1 (page 2, lines 86-94). The experimental results in Section 3.4 (page 15-20) show that users can not only understand the specific meaning of the features (in this paper, edge-like features, with further research to be conducted) but also benefit from reduced computational complexity and higher classification accuracy using this method.

Since our method differs significantly from CNNs, which may pose challenges to comprehension. The article is structured according to the methodological workflow sequence, with interpretability integrated into each stage. Below is a description that consolidates the interpretability aspect, hoping to clearly express the authors' thought process:

Most neural network feature extraction models, such as CNNs, output features from input images that lack interpretability, meaning humans cannot understand these features, which is why they are often referred to as black-box models. This is disadvantageous for tasks that require an understanding of the underlying principles or tasks where safety and certainty are especially important. KAN, on the other hand, is interpretable, as it can calculate the contribution of each node to the network, select key pixels with significant contributions, and visualize them for human or mathematical interpretation. In this paper, we leverage this feature of KAN and use a modified KAN interpretability technique to identify edge-like pixels in the image as key points. This approach has two advantages: first, it helps us understand which points in the image are useful for subsequent tasks, and second, edge-like points can be directly extracted, rather than using neural networks for extraction. This significantly improves computational efficiency, as the extracted pixels, when used as input to KAN, reduce the input dimension by over 80% compared to the original image, thereby enhancing the computational efficiency of the subsequent network.

Comments 2: compare the obtained results with other existing ones in order to show the benefits of the proposed method

Response 2: Agree. In Section 3, we have compared the proposed method, PFSKAN, with existing CNN methods. Section 3.1 (page 13, line 326) provides a detailed description of the comparison models, where ConvMLP is the existing CNN method, and ConvKAN is a combination of CNN and KAN, also an existing approach, introduced for a better comparison with PFSKAN to demonstrate the effectiveness of PFS. The experimental results in Section 3.4 (page 15-20) present the comparison outcomes of these models, showing that the proposed method outperforms others in terms of accuracy and parameter efficiency.

Comments 3: how can be applied the proposed method for other datasets (on the same task - classification)

Response 3: Thank you for pointing this out. The purple section on the right side of Figure 1 (page 4) represents the application on other datasets. As we have demonstrated in this paper, edge-like pixels are key feature points. Therefore, for other datasets, we only need to use the EdgeFuseTrim detection method to extract the feature points from the dataset as input to the KAN network. The subsequent classification task is still completed by training the KAN model, as shown in the application on Fashion-MNIST (page 16, line 443) and CIFAR-100_10 (page 19, line 500) in Section 3. However, this application is only suitable for datasets with similar resolutions, and higher resolution datasets are still under further investigation. A limitations analysis is provided in the last paragraph of Section 3.4 (page 18, line 487-496) and in line 524-527 of Section 4 (page 20).

Round 2

Reviewer 1 Report

Comments and Suggestions for Authors
  • The manuscript is revised based on the reviewers' comments.  
  • Readers will have a clearer understanding of the results of the comparative performance analysis. 
  • The authors now mention their limitations and future goals.

Reviewer 2 Report

Comments and Suggestions for Authors

The paper can be accepted at current stage.